# Reconstruction of Type II Supergravities via $O(d) \times O(d)$ Duality Invariants

**Yoshifumi Hyakutake *** and **Kiyoto Maeyama**

College of Science, Ibaraki University, Bunkyo 2-1-1, Mito 310-8512, Ibaraki, Japan; 21nd103y@vc.ibaraki.ac.jp
* Correspondence: yoshifumi.hyakutake.phys@vc.ibaraki.ac.jp

**Abstract:** We reconstruct type II supergravities by using building blocks of $O(d) \times O(d)$ invariants. These invariants are obtained by explicitly analyzing $O(d) \times O(d)$ transformations of 10 dimensional massless fields. Similar constructions are performed by employing double field theory or generalized geometry, but we completed the reconstruction within the framework of the supergravities.

**Keywords:** string theory; supergravity; duality





## 1. Introduction

Dualities among superstring theories play important roles to reveal both perturbative and non-perturbative aspects of superstring theories. Especially, type IIA superstring theory is related to type IIB superstring theory by T-duality, which interchanges Kaluza–Klein modes (KK modes) and winding modes of a compactified circle direction [1,2]. In the low energy limit, massive modes in the type II superstring theories are decoupled, and the effective actions are well described by corresponding type II supergravity theories [3,4]. The T-duality transformations of background massless fields are well known as Buscher rule [5,6].

When the superstring theories are toroidally compactified on $T^d$, the duality transformation can be generalized to $O(d,d)$ duality [7,8]. Actually it is argued in ref. [9] that, by assuming all fields depend only on a time coordinate, NS-NS sector in the low energy effective action, which consists of a graviton, a dilaton and Kalb–Ramond B field (B field), can be rewritten in manifestly $O(d,d)$ invariant expression. In addition, $O(d,d)$ invariance of the NS-NS sector in general background was confirmed in refs. [10,11]. Furthermore, it is also proven that $O(d,d)$ invariance can be extended to all orders in $\alpha'$ corrections to the low energy effective action [12].

$O(d,d)$ transformation of the R-R sector has been investigated in refs. [13–20]. One approach is to note that R-R potentials fill up a spinor representation of $SO(d,d)$ duality transformation [13,14]. The spinor representation of R-R potentials combined with B field was explicitly constructed when the compactified space was $T^3$ [15], and a general case of $T^d$ compactification was completed in ref. [16]. Another approach was investigated by Hassan in refs. [17–20], where the consistency of the duality transformation with local supersymmetry transformation is imposed. In this approach, the $O(d,d)$ transformations of dilatinos and gravitinos are explicitly written in terms of 10 dimensional forms, and those of R-R potentials are derived in a bispinor form. In the type II superstring theories, the formulation of superspace that is compatible with T-duality was discussed in ref. [21], and inclusion of R-R fields and an application to AdS background were investigated in refs. [22,23]. Generalization of ref. [20] to non-abelian T-duality was performed in refs. [24,25].

Although the type II supergravities possess $O(d,d)$ duality invariance, forms of the action are not manifestly invariant in terms of 10 dimensional fields. There are two formalisms to improve this point. The first one is a double field theory, which treats internal

coordinates of winding modes and KK modes simultaneously [26–28]. $O(d, d)$ transformation is realized as a rotation among these $2d$ coordinates, and fields are generalized to behave as tensors under this coordinate transformation. $O(d, d)$ invariant forms of the type II supergravities are discussed in the framework of the double field theory in refs. [29–31]. The second one is a generalized geometry, which treats tangent and cotangent bundles of compactified manifold on equal footing [32–34]. Lie brackets of two vector fields are also modified to Courant brackets to incorporate B field transformation with the general coordinate one. $O(d, d)$ invariant forms of the type II supergravities are discussed in the framework of the generalized geometry in ref. [35].

The double field theory or the generalized geometry played important roles to reveal the $O(d, d)$ invariant structure, however, it is not so clear to derive such structure within the framework of the type II supergravities. In this paper, we revisit the $O(d) \times O(d)$ subgroup of the duality transformation discussed in ref. [20] to construct $O(d) \times O(d)$ invariants within the framework of the type II supergravities. We review that $O(d) \times O(d)$ transformations of NS-NS fields and fermionic fields are completely written in terms of 10 dimensional fields and construct $O(d) \times O(d)$ invariants by evaluating these. The actions of the type II supergravities are completely written by combinations of these building blocks, which are consistent with ones obtained in refs. [31,35].

This paper is organized as follows. In Section 2, we review the $O(d) \times O(d)$ duality transformations of fields shown in ref. [20]. Especially, we show that these transformations can be written by using 10 dimensional fields[1]. In Section 3, we construct $O(d) \times O(d)$ duality invariants for NS-NS fields and fermionic ones. We also check these duality invariants in the background of fundamental strings and wave solutions, or NS5-branes and KK monopoles. In Section 4, we construct NS-NS bosonic terms in the type II supergravities by using the duality invariants. In Section 5, we construct fermionic bilinear terms in the type II supergravities by duality invariants. Section 6 is devoted to conclusions and discussions. In Appendix A, we review the actions of the type II supergravities for the NS-NS sector and fermionic bilinear terms.

## 2. Brief Review of $O(d) \times O(d)$ Transformations

In this section, we briefly review $O(d) \times O(d)$ transformations of massless fields in the type II supergravities. NS-NS fields of the type II supergravities consist of the graviton $G_{MN}$, the Kalb–Ramon field $B_{MN}$ and the dilaton $\Phi$. First, we take into account these fields to show a standard dimensional reduction of 10 dimensional supergravity action to $10 - d$ non-compact dimensions [11]. The reduced action is written in a manifestly $O(d, d)$ invariant form, and $O(d, d)$ transformations of reduced fields can be obtained by using $2d \times 2d$ matrix notation. Among $O(d, d)$ transformations of NS-NS fields, $O(d) \times O(d)/O(d)$ transformations are non-trivial if we ignore general linear coordinate transformations and shift of the B field [12]. Thus, we focus on the transformations of $O(d) \times O(d)$, and it is possible to express the duality transformations in terms of the original 10 dimensional fields. Then, we review the $O(d) \times O(d)$ transformations of fermionic fields, two gravitinos and two dilatinos, which are compatible with local supersymmetry transformations in the type II supergravities.

We denote the 10 dimensional spacetime indices as $K, L, M, N, \cdots$. Non-compact spacetime directions are labeled by $\mu, \nu, \cdots$ and compact $d$ dimensions are done by $\alpha, \beta, \cdots$. On the other hand, local Lorentz indices are denoted as $A, B, C, D, \cdots$. Non-compact local Lorentz indices are labeled by $i, j, \cdots$, and those for compact $d$ dimensions are noted by $a, b, \cdots$. The explanation in this section is based on ref. [20], but some of the transformations are not written in 10 dimensional fields there, which are repaired below.

The bosonic part of the action for NS-NS fields is common to both type II supergravities, and the explicit form is written as

$$S_{10} = \frac{1}{2\kappa_{10}^2} \int d^{10}x \sqrt{-G}\, e^{-2\Phi} \left( R + 4\partial_M \Phi \partial^M \Phi - \frac{1}{2 \cdot 3!} H_{MNP} H^{MNP} \right), \tag{1}$$

where $M, N, P = 0, 1, \cdots, 9$ and $\kappa_{10}^2$ is the gravitational constant in 10 dimensions. $R$ is a scalar curvature and $H_{MNP} = 3\partial_{[M}B_{NP]}$ is a three-form field strength of the B field. Now, we consider the dimensional reduction of the above action on $d$ dimensional torus. The dimensional reduction of the metric is given by

$$G_{MN} = \begin{pmatrix} g_{\mu\nu} + g_{\gamma\delta}A_\mu^\gamma A_\nu^\delta & -g_{\beta\gamma}A_\mu^\gamma \\ -g_{\alpha\gamma}A_\nu^\gamma & g_{\alpha\beta} \end{pmatrix}, \tag{2}$$

where $\mu, \nu = 0, 1, \cdots, 9 - d$ and $\alpha, \beta, \gamma, \delta = 10 - d, \cdots, 9$. Here, $g_{\mu\nu}$ is a metric, $A_\mu^\gamma$ are $U(1)$ gauge fields and $g_{\alpha\beta}$ are scalars for non-compact spacetime directions. Note that all fields are assumed to be dependent on $x^\mu$ directions but not on $x^\alpha$ ones. The dimensional reduction of the three-form field strength is a little bit complicated, and it is easier to consider in the local Lorentz frame. By using a vielbein $E^M{}_A$ in 10 dimensions, the three-form field in the local Lorentz frame is defined as $H_{ABC} = E^M{}_A E^N{}_B E^P{}_C H_{MNP}$, and the dimensional reduction of each component is written as

$$
\begin{aligned}
H_{ijk} &= e^\mu{}_i e^\nu{}_j e^\rho{}_k \left( h_{\mu\nu\rho} - \frac{3}{2}\tilde{A}_{\alpha[\rho}F_{\mu\nu]}^\alpha - \frac{3}{2}\tilde{F}_{\alpha[\mu\nu}A_{\rho]}^\alpha \right), \\
H_{ija} &= e^\mu{}_i e^\nu{}_j e^\alpha{}_a \left( -\tilde{F}_{\alpha\mu\nu} + B_{\alpha\beta}F_{\mu\nu}^\beta \right), \\
H_{iab} &= e^\mu{}_i e^\alpha{}_a e^\beta{}_b \partial_\mu B_{\alpha\beta}.
\end{aligned}
\tag{3}
$$

Note that $H_{abc} = 0$ since all fields are dependent only on $x^\mu$. Here, $F_{\mu\nu}^\alpha = 2\partial_{[\mu}A_{\nu]}^\alpha$ are gauge field strengths. Gauge fields, which originate from B field, are defined as $\tilde{A}_{\alpha\mu} = B_{\alpha\mu} + B_{\alpha\beta}A_\mu^\beta$, and $\tilde{F}_{\alpha\mu\nu} = \partial_\mu\tilde{A}_{\alpha\nu} - \partial_\nu\tilde{A}_{\alpha\mu}$ are corresponding gauge field strengths. The dimensional reduction of the dilaton field is defined as

$$\sqrt{\det(g_{\alpha\beta})}e^{-2\Phi} = e^{-2\phi}. \tag{4}$$

$\phi$ is a dilaton field in non-compact directions. Substituting Equations (2)–(4) into the 10-dimensional action (1), we obtain the action for the non-compact directions of the form

$$
\begin{aligned}
S_{10-d} = \frac{V_d}{2\kappa_{10}^2} \int d^{10-d}x \sqrt{-\det(g_{\mu\nu})}\, e^{-2\phi} \Bigg[ & r + 4\partial_i\phi\partial^i\phi \\
& -\frac{1}{8}\mathrm{Tr}\left( \hat{\eta}\,\partial_i\mathcal{H}\,\hat{\eta}\,\partial^i\mathcal{H} \right) - \frac{1}{4}\begin{pmatrix} F_{ij} & \tilde{F}_{ij} \end{pmatrix}\hat{\eta}\mathcal{H}\hat{\eta}\begin{pmatrix} F^{ij} \\ \tilde{F}^{ij} \end{pmatrix} - \frac{1}{2\cdot 3!}H_{ijk}H^{ijk} \Bigg],
\end{aligned}
\tag{5}
$$

where $V_d$ is a volume of the $d$ dimensional torus and $r$ is a scalar curvature constructed out of $g_{\mu\nu}$. The indices for the compact directions are expressed by the matrix notation, as will be explained below.

Since $O(d, d)$ transformations act on indices for compactified directions, fields only with non-compact directions, $g_{\mu\nu}$, $\phi$ and $h_{\mu\nu\rho}$, are invariant under $O(d, d)$ transformations. The first line of action (5) consists of kinetic terms of $g_{\mu\nu}$ and $\phi$, so this line is invariant under $O(d, d)$ transformation. In the second line, scalar fields with compact spatial indices $g_{\alpha\beta}$ and $B_{\alpha\beta}$ are gathered into

$$\mathcal{H} = \begin{pmatrix} g^{-1} & -g^{-1}B \\ Bg^{-1} & g - Bg^{-1}B \end{pmatrix}, \tag{6}$$

and the $O(d, d)$ transformation $\mathcal{O}$ for massless NS-NS fields is defined by [9]

$$\mathcal{H}' = \mathcal{O}\mathcal{H}\mathcal{O}^T, \qquad \begin{pmatrix} A_i' \\ \tilde{A}_i' \end{pmatrix} = \mathcal{O}\begin{pmatrix} A_i \\ \tilde{A}_i \end{pmatrix}, \qquad \mathcal{O}^T\hat{\eta}\,\mathcal{O} = \hat{\eta}, \qquad \hat{\eta} = \begin{pmatrix} 0 & \mathbf{1}_d \\ \mathbf{1}_d & 0 \end{pmatrix}. \tag{7}$$

Here, $\mathcal{H}$ and $\mathcal{O}$ are $2d \times 2d$ matrices and $\mathcal{O}$ acts on the gauge indices of $A_i^\alpha$ and $\tilde{A}_{\alpha i}$. It is obvious that the first and second terms in the second line of Equation (5) are invariant under these transformations. As for the third term in the second line, $H_{ijk}$ contains $A_\mu^\alpha$, $\tilde{A}_{\alpha\mu}$ and their field strengths, which transform under $O(d,d)$ transformation, but still $H_{ijk}$ is $O(d,d)$ invariant.

There are $d(2d-1)$ elements for the duality transformations of $O(d,d)$, but some of them are trivial in the sense that these do not mix NS-NS fields. Actually, $d^2$ elements of general linear coordinate transformation $GL(d)$ and $\frac{1}{2}d(d-1)$ elements for the shift of the B field are trivial. The remaining $\frac{1}{2}d(d-1)$ elements are non-trivial, and these construct a subgroup of $O(d) \times O(d)/O(d)$. The $O(d) \times O(d)$ subgroup is expressed as [12]

$$\mathcal{O} = \frac{1}{2}\begin{pmatrix} \mathcal{S} + \mathcal{R} & \mathcal{S} - \mathcal{R} \\ \mathcal{S} - \mathcal{R} & \mathcal{S} + \mathcal{R} \end{pmatrix}, \qquad \mathcal{S}^T\mathcal{S} = \mathcal{R}^T\mathcal{R} = \mathbf{1}_d. \tag{8}$$

The case of $\mathcal{S} = \mathcal{R}$ corresponds to a part of general linear coordinate transformation.

From Equation (7), it is possible to extract duality transformations of dimensionally reduced fields. These are then gathered into duality transformations of the original 10 dimensional fields. Below, we summarize $O(d) \times O(d)$ transformations of fields in 10 dimensions [18]. By introducing $10 \times 10$ matrices as

$$Q_\pm = \frac{1}{2}(S+R) \mp \frac{1}{2}(S-R)(G \mp B), \tag{9}$$

$$S = \begin{pmatrix} \mathbf{1}_{10-d} & 0 \\ 0 & \mathcal{S} \end{pmatrix}, \qquad R = \begin{pmatrix} \mathbf{1}_{10-d} & 0 \\ 0 & \mathcal{R} \end{pmatrix},$$

the 10-dimensional inverse metric transforms as

$$G'^{-1} = Q_\pm G^{-1} Q_\pm^T. \tag{10}$$

Since the duality invariant, which includes the dilaton field, is written by $\Phi - \frac{1}{4}\log \det G$, the duality transformation of the dilaton field is given by

$$\Phi' = \Phi - \frac{1}{2}\log \det Q_\pm. \tag{11}$$

The $\pm$ sign originates from actions to the world-sheet left and right moving modes, respectively. From Equation (10), it is possible to define $O(d) \times O(d)$ transformation of the vielbein as

$$E'^M_{(\pm)A} = Q^M_{\pm N} E^N{}_A. \tag{12}$$

Notice that $E'^M_{(\pm)A}$ are related by local Lorenz transformation of

$$E'^M_{(+)A} = E'^M_{(-)B} \Lambda^B{}_A, \qquad \Lambda^B{}_A = E^B{}_M Q_-^{-1M}{}_N Q^N_{+K} E^K{}_A. \tag{13}$$

Thus, the local Lorentz frame of the left moving sector is obtained by twisting that of the right moving sector by $\Lambda^A{}_B$. Therefore, invariants under local Lorentz transformation, which are constructed out of $E'^M_{(+)A}$, can always be written in terms of $E'^M_{(-)A}$.

Since two kinds of vielbein can be used after the duality transformation, the three-form field strength $H_{ABC} = E^M{}_A E^N{}_B E^K{}_C H_{MNK}$ also transforms in two ways as [20]

$$H'_{(\pm)ABC} = E'^M_{(\pm)A} E'^N_{(\pm)B} E'^K_{(\pm)C} H'_{MNK}$$
$$= H_{ABC} - 3G_{KM} Q_\pm^{-1M}{}_N (S-R)^{NL} W^\pm_{L[BC} E^K{}_{A]}. \tag{14}$$

Here, $W_M^{\pm A}{}_B$ are connections defined by using torsionless spin connection $\Omega_M{}^A{}_B$ as

$$W_M^{\pm A}{}_B = \Omega_M{}^A{}_B \mp \frac{1}{2} H_M{}^A{}_B, \tag{15}$$

and the duality transformations are calculated as

$$W'^{\pm}_{(\pm)M}{}^A{}_B = W_N^{\pm A}{}_B Q_{\mp}^{-1N}{}_M. \tag{16}$$

Notice that $W'^{\pm}_{(\pm)M}{}^A{}_B$ are constructed out of $E'^M_{(\pm)A}$, respectively. Similarly, $\Gamma^{\pm K}{}_{MN}$ are connections defined by using affine connection $\Gamma^K{}_{MN}$ as

$$\Gamma^{\pm K}{}_{MN} = \Gamma^K{}_{MN} \pm \frac{1}{2} H^K{}_{MN}, \tag{17}$$

and the duality transformations are derived as

$$\Gamma'^{\pm K}{}_{MN} = Q_{\pm K'}^K \Gamma^{\pm K'}{}_{M'N'} Q_{\mp}^{-1M'}{}_M Q_{\pm}^{-1N'}{}_N - \partial_M(Q_{\pm L}^K) Q_{\pm}^{-1L}{}_N. \tag{18}$$

Since the vielbein is not used in Equation (17), there are no ($\pm$) subscriptions in the above.

Next, let us summarize duality transformations of gravitinos $\Psi_{\pm M}$ and dilatinos $\lambda_{\pm}$. In ref. [20], these transformations are derived so as to be consistent with the local supersymmetry (A4). It is easy to check this for the gravitino $\Psi_{-M}$, and the result is

$$\Psi'_{-M} = \Psi_{-N} Q_+^{-1N}{}_M, \qquad \epsilon'_- = \epsilon_-. \tag{19}$$

To derive the above, we used $Q_{\pm M}^N \partial_N = \partial_M$. This holds because the derivatives of fields with respect to the compact directions are zero. For the gravitino $\Psi_{+M}$, the duality transformation $\Psi'_{+M}$ is defined by using $E'^M_{(-)A}$ and the susy transformation becomes

$$\begin{aligned}
\delta_+ \Psi'_{+M} &= 2\left(\partial_M + \frac{1}{4} W'^+_{(-)MAB} \Gamma^{AB}\right)\epsilon'_+ + \cdots \\
&= 2U_+\left(\partial_M + \frac{1}{4} W'^+_{(+)MAB} \Gamma^{AB}\right)U_+^{-1}\epsilon'_+ + \cdots, \tag{20}
\end{aligned}$$

where $\Gamma^A$ is a gamma matrix in 10 dimensions. In the above, we ignored R-R fields and used local Lorentz transformation to change $E'^M_{(-)A}$ to $E'^M_{(+)A}$. $U_+$ is a spinor representation of the local Lorentz transformation of $\Lambda^{-1}$ and satisfies $U_+ \Gamma^A U_+^{-1} = \Lambda^{-1A}{}_B \Gamma^B$. Equation (20) is compatible with the duality transformation if we define

$$\Psi'_{+M} = U_+ \Psi_{+N} Q_-^{-1N}{}_M, \qquad \epsilon'_+ = U_+ \epsilon_+. \tag{21}$$

Finally, we consider $O(d) \times O(d)$ duality transformations of dilatinos. As in the case of the gravitinos, the duality transformations are derived so as to be consistent with the local supersymmetry (A4).

$$\begin{aligned}
\delta_- \lambda'_- &= 2\left(\Gamma'^M_{(-)}\partial_M \Phi' + \frac{1}{12}\Gamma^{ABC} H'_{(-)ABC}\right)\epsilon'_- + \cdots \\
&= 2\left\{\Gamma^M \partial_M \Phi - \frac{1}{2}\Gamma^B E^A{}_M Q_-^{-1M}{}_N (S-R)^{NL} W^-_{LAB}\right. \\
&\quad \left. + \frac{1}{12}\Gamma^{ABC} H_{ABC} - \frac{1}{4}\Gamma^{ABC} E_{CM} Q_-^{-1M}{}_N (S-R)^{NL} W^-_{LAB}\right\}\epsilon_- + \cdots \\
&= 2\left(\Gamma^M \partial_M \Phi + \frac{1}{12}\Gamma^{ABC} H_{ABC}\right)\epsilon_- - \frac{1}{2}\Gamma^C \Gamma^{AB} E_{CM} Q_-^{-1M}{}_N (S-R)^{NL} W^-_{LAB}\epsilon_- + \cdots \\
&= \delta_-\left(\lambda_- - Q_-^{-1M}{}_N (S-R)^{NL}\Gamma_M \Psi_{-L}\right). \tag{22}
\end{aligned}$$

In the second equality, we used Equation (14) and employed the fifth line of Equation (32). Thus, the duality transformation of the dilatino $\lambda_-$ is compatible with the local supersymmetry if we define

$$\lambda'_- = \lambda_- - Q_-^{-1M}{}_N (S-R)^{NL} \Gamma_M \Psi_{-L}. \tag{23}$$

As in the case of the gravitino, the duality transformation $\lambda'_+$ is defined by using $E'^M_{(-)A}$. By taking into account the local Lorentz transformation, we obtain

$$\lambda'_+ = U_+ \left( \lambda_+ + Q_+^{-1M}{}_N (S-R)^{NL} \Gamma_M \Psi_{+L} \right). \tag{24}$$

## 3. $O(d) \times O(d)$ Duality Invariants

In this section, we construct $O(d) \times O(d)$ duality invariants. In order to find these, let us prepare useful relations for $Q^M_{\pm N}$. $Q_\pm$ are defined in the $10 \times 10$ matrix notation as (9), and by noting $S^T S = R^T R = 1$, we obtain

$$
\begin{aligned}
Q_\pm (S-R)^T &= \frac{1}{2}(S+R)(S-R)^T \mp \frac{1}{2}(S-R)(G \mp B)(S-R)^T \\
&= -(S-R)\left\{ \frac{1}{2}(S+R) \pm \frac{1}{2}(S-R)(G \pm B) \right\}^T \\
&= -(S-R)Q_\mp^T.
\end{aligned}
\tag{25}
$$

It is often useful to express the above as follows.

$$Q_\pm^{-1}(S-R) = -\left\{ Q_\mp^{-1}(S-R) \right\}^T. \tag{26}$$

On the other hand, from Equation (9), $Q_+$ is written by $Q_-$ as

$$Q_+ = Q_- - (S-R)G. \tag{27}$$

By multiplying $Q_\mp^{-1}$ from the left, we find

$$Q_\mp^{-1}(S-R)G = \pm(1 - Q_\mp^{-1}Q_\pm). \tag{28}$$

Combining Equations (26) and (28), we obtain a useful relation

$$
\begin{aligned}
GQ_\pm^{-1}(S-R) &= -\left\{ Q_\mp^{-1}(S-R)G \right\}^T \\
&= \mp 1 \pm \left\{ Q_\mp^{-1}Q_\pm \right\}^T.
\end{aligned}
\tag{29}
$$

This relation is often used to construct $O(d) \times O(d)$ duality invariants.

### 3.1. Duality Invariants $S^\pm_{ABC}$ and $T^\pm_A$ for NS-NS Bosonic Fields

Now, we construct duality invariants for NS-NS bosonic fields. $O(d) \times O(d)$ transformation of $H_{ABC}$ in 10 dimensions is evaluated as follows.

$$
\begin{aligned}
H'_{(\pm)ABC} &= H_{ABC} - 3G_{KM}Q_\pm^{-1M}{}_N (S-R)^{NL} W^\pm_{L[BC} E^K{}_{A]} \\
&= H_{ABC} \pm 3W^\pm_{[ABC]} \mp 3Q_\mp^{-1L}{}_N Q^N_{\pm K} W^\pm_{L[BC} E^K{}_{A]} \\
&= H_{ABC} \pm 3W^\pm_{[ABC]} \mp 3W'^\pm_{(\pm)[ABC]},
\end{aligned}
\tag{30}
$$

where $W^\pm_{ABC} = E^M{}_A W^\pm_{MBC}$ and $W'^\pm_{ABC} = E'^M_{(\pm)A} W'^\pm_{(\pm)MBC}$. In the second line, we used Equation (29). Thus, we find $O(d) \times O(d)$ duality invariant of the form

$$S^\pm_{ABC} \equiv H_{ABC} \pm 3W^\pm_{[ABC]} = -\frac{1}{2}H_{ABC} \pm 3\Omega_{[ABC]}. \tag{31}$$

Note that these do not behave as tensors under general coordinate transformation and $E'^M_{(+)A}$ is used for the $+$ mode of the dual theory. This means that $S'^\pm_{(\pm)ABC} = S^\pm_{ABC}$.

$O(d) \times O(d)$ transformation of the dilaton is given by Equation (11), and the derivative of that equation is evaluated as

$$
\begin{aligned}
\partial_\mu \Phi' - \partial_\mu \Phi &= -\frac{1}{2}\partial_\mu \log \det Q_\pm \\
&= -\frac{1}{2}Q_\pm^{-1\alpha}{}_\beta \partial_\mu Q_\pm{}^\beta{}_\alpha \\
&= \pm\frac{1}{4}Q_\pm^{-1\alpha}{}_\beta (S-R)^{\beta\gamma}\partial_\mu (G\mp B)_{\gamma\alpha} \\
&= \pm\frac{1}{2}e^a{}_\alpha Q_\pm^{-1\alpha}{}_\beta (S-R)^{\beta\gamma}W^\pm_{\gamma ai}e^i{}_\mu \\
&= \pm\frac{1}{2}E^A{}_M Q_\pm^{-1M}{}_N (S-R)^{NL}W^\pm_{LAB}E^B{}_\mu \\
&= -\frac{1}{2}E^{MA}W^\pm_{MAB}E^B{}_\mu + \frac{1}{2}E^{MA}Q_\mp^{-1L}{}_N Q^N_{\pm M}W^\pm_{LAB}E^B{}_\mu \\
&= -\frac{1}{2}E^{MA}W^\pm_{MAB}E^B{}_\mu + \frac{1}{2}E'^{MA}_{(\pm)}W'^\pm_{(\pm)MAB}E'^B_{(\pm)\mu}.
\end{aligned}
\tag{32}
$$

Equation (29) is used in the sixth line, and $E'^i_{(\pm)\mu} = E^i_{(\pm)\mu}$ is used in the last line. Since $\partial_\alpha \Phi = 0$ and $E^{MA}W^\pm_{MAB}E^B{}_\alpha = 0$, we find $O(d) \times O(d)$ invariant of the form

$$
T^\pm_N \equiv \partial_N \Phi - \frac{1}{2}W^{\pm A}{}_{AN} = \partial_N \Phi - \frac{1}{2}\Omega^A{}_{AN},
\tag{33}
$$

where $W^{\pm A}{}_{AN} = E^{MA}W^\pm_{MAB}E^B{}_N$. Notice that in the dual theory, $E'^M_{(+)A}$ is used for the $+$ mode, so we obtained $T'^\pm_{(\pm)N} = T^\pm_N$. Since $T^\pm_M Q^M_{\pm N} = T^\pm_N$ holds, $T^\pm_A = E^M{}_A T^\pm_M$ is also $O(d) \times O(d)$ invariant. Invariants thar are similar to $S^\pm_{ABC}$ and $T^\pm_A$ are also constructed in the flux formulation of the double field theory [36].

### 3.2. Duality Invariants $\Theta_\pm$ for Fermionic Fields

$O(d) \times O(d)$ transformations of the dilatinos in 10 dimensions are given by

$$
\begin{aligned}
\lambda'_\pm &= U_\pm \left\{ \lambda_\pm \pm G_{KM}Q_\pm^{-1M}{}_N (S-R)^{NL}E^K{}_A \Gamma^A \Psi_{\pm L} \right\} \\
&= U_\pm \left\{ \lambda_\pm - E^M{}_A \Gamma^A \Psi_{\pm M} + Q_\mp^{-1L}{}_N Q^N_{\pm M}E^M{}_A \Gamma^A \Psi_{\pm L} \right\} \\
&= U_\pm \left\{ \lambda_\pm - E^M{}_A \Gamma^A \Psi_{\pm M} + E'^M_{(\pm)A}\Gamma^A U_\pm^{-1}\Psi'_{\pm M} \right\} \\
&= U_\pm \left\{ \lambda_\pm - E^M{}_A \Gamma^A \Psi_{\pm M} \right\} + E'^M_{(-)A}\Gamma^A \Psi'_{\pm M},
\end{aligned}
\tag{34}
$$

where $U_- = 1$ and $U_+$ is a spinor representation of local Lorentz transformation whose corresponding vector representation is given by $\Lambda^A{}_B = E^A{}_M Q_-^{-1M}{}_N Q^N_{+L}E^L{}_B$. We used Equation (29) in the second line and $U_+ \Gamma^A U_+^{-1} = \Lambda^{-1A}{}_B \Gamma^B$ in the fourth line. Thus, we find duality invariants up to local Lorentz transformation $U_\pm$.

$$
\Theta_\pm = \lambda_\pm - E^M{}_A \Gamma^A \Psi_{\pm M}.
\tag{35}
$$

Notice that the dual theory is written by $E'^M_{(-)A}$ for $\Theta_\pm$. This means $\Theta'_{(-)\pm} = U_\pm \Theta_\pm$, which is different from $S^\pm_{ABC}$ and $T^\pm_A$. Similar expressions to the above are also obtained in the framework of the double field theory [31] or generalized geometry [35].

### 3.3. Check of Duality Invariants for Classical Solutions

Since we have constructed $O(d) \times O(d)$ invariants, let us evaluate these values for classical solutions that exchange under T-duality. We examine two pairs of solutions in the type II supergravities. The first one is fundamental strings stretching along the $X^9$ direction

and waves propagating along the same direction. The fundamental strings carry charges with respect to B field, and $B_{09}$ becomes non-trivial. On the other hand, the wave solution has a non-trivial component of $G_{09}$. It is well known that these two solutions are exchanged under Buscher rule. The second pair is smeared NS5-branes and KK monopoles. The NS5-branes are stretching along $X^1, \cdots, X^5$ directions and smeared along the $X^9$ direction. Thus, the smeared NS5-branes are localized at the origin of $(X^6, X^7, X^8)$ directions. On the other hand, if we compactify the type II supergravities along the $X^9$ direction, there is a $U(1)$ gauge field in the dimensionally reduced theory. KK monopoles are magnetically charged with respect to this $U(1)$ gauge field and localized at the origin of $(X^6, X^7, X^8)$ directions. It is also well known that these two solutions are exchanged under Buscher rule.

First, we consider classical solutions of fundamental strings and waves. The solution of the fundamental strings is given by

$$ds^2 = -h_1^{-1}(dX^0)^2 + \sum_{i=1}^{8}(dX^i)^2 + h_1^{-1}(dX^9)^2, \tag{36}$$

$$e^{\Phi} = h_1^{-\frac{1}{2}}, \quad B_{09} = -1 + h_1^{-1}, \quad h_1 = 1 + \frac{c_1}{r^6},$$

where $r^2 = \sum_{i=1}^{8}(X^i)^2$. Moreover, non-trivial components of $O(d) \times O(d)$ invariants for this solution are evaluated as

$$S^{\pm}_{\hat{i}\hat{0}\hat{9}} = \frac{1}{2}h_1^{-1}\partial_i h_1. \tag{37}$$

On the other hand, the dual solution of the wave along the $X^9$ direction is given by

$$ds^2 = -h_{\rm w}^{-1}(dX^0)^2 + \sum_{i=1}^{8}(dX^i)^2 + h_{\rm w}\big(dX^9 - (1 - h_{\rm w}^{-1})dX^0\big)^2, \quad h_{\rm w} = 1 + \frac{c_{\rm w}}{r^6}. \tag{38}$$

And non-trivial components of $O(d) \times O(d)$ invariants for this solution are calculated as

$$S'^{\pm}_{(-)\hat{i}\hat{0}\hat{9}} = \mp\frac{1}{2}h_{\rm w}^{-1}\partial_i h_{\rm w}. \tag{39}$$

Here, we used hats for local Lorentz indices. If we set $h_1 = h_{\rm w}$, we obtain $S^{\pm}_{ABC} = S'^{\pm}_{(\pm)ABC}$ because $E'^M_{(-)A} = E'^M_{(+)A}$ except for $E'^9_{(-)\hat{9}} = -E'^9_{(+)\hat{9}}$.

Second, we consider classical solutions of smeared NS5-branes and KK monopoles. The solution of the NS5-branes smeared along the $X^9$ direction is given by

$$ds^2 = -(dX^0)^2 + (dX^1)^2 + \cdots + (dX^5)^2 + h_5 \sum_{i=6,7,8}(dX^i)^2 + h_5(dX^9)^2, \tag{40}$$

$$e^{\Phi} = h_5^{\frac{1}{2}}, \quad H_{ij9} = \epsilon_{ijk}\partial^k h_5, \quad h_5 = 1 + \frac{c_5}{r^6},$$

where $r^2 = \sum_{i=6,7,8}(X^i)^2$. Then, non-trivial components of $O(d) \times O(d)$ invariants for this solution are evaluated as

$$S^{\pm}_{\hat{i}\hat{j}\hat{9}} = -\frac{1}{2}h_5^{-\frac{3}{2}}\epsilon_{ijk}\partial^k h_5, \quad T_{\hat{i}} = -\frac{1}{4}h_5^{-\frac{3}{2}}\partial_i h_5. \tag{41}$$

On the other hand, the dual solution of the KK monopoles is given by

$$ds^2 = -(dX^0)^2 + (dX^1)^2 + \cdots + (dX^5)^2 + h_{\rm m} \sum_{i=6,7,8}(dX^i)^2 + h_{\rm m}^{-1}\big(dX^9 - A_i dX^i\big)^2, \tag{42}$$

$$F_{ij} = \partial_i A_j - \partial_j A_i = -\epsilon_{ijk}\partial^k h_{\rm m}, \quad h_{\rm m} = 1 + \frac{c_{\rm m}}{r^6}.$$

And non-trivial components of $O(d) \times O(d)$ invariants for this solution become

$$S'^{\pm}_{(-)\hat{i}\hat{j}\hat{9}} = \pm\frac{1}{2}h_{\mathrm{m}}^{-\frac{3}{2}}\epsilon_{ijk}\partial^k h_{\mathrm{m}}, \quad T'_{(-)\hat{i}} = -\frac{1}{4}h_{\mathrm{m}}^{-\frac{3}{2}}\partial_i h_{\mathrm{m}}. \tag{43}$$

Here, we used hats for local Lorentz indices. If we set $h_5 = h_{\mathrm{m}}$, we obtain $S^{\pm}_{ABC} = S'^{\pm}_{(\pm)ABC}$ because $E'^M_{(-)A} = E'^M_{(+)A}$ except for $E'^9_{(-)\hat{9}} = -E'^9_{(+)\hat{9}}$.

## 4. Construction of NS-NS Bosonic Terms in Type II Supergravity via Duality Invariants

Let us construct NS-NS bosonic terms in the type II supergravities by using duality invariants. Building blocks are $S^{\pm}_{ABC}$, $T_A$ and $W^{\pm}_{MAB}$. The action consists of two derivative terms, so candidates are $S^{\pm}_{ABC}S^{\pm ABC}$, $T_A T^A$ and $G^{MN}W^{\pm}_{MAB}W^{\pm AB}_N = W^{\pm ABC}W^{\pm}_{ABC}$ multiplied by $Ee^{-2\Phi}$.

First, we evaluate $Ee^{-2\Phi}S^{\pm}_{ABC}S^{\pm ABC}$.

$$\begin{aligned}
&Ee^{-2\Phi}S^{\pm}_{ABC}S^{\pm ABC} \\
&= Ee^{-2\Phi}\left\{H_{ABC}H^{ABC} \pm 6H^{ABC}W^{\pm}_{ABC} + 9W^{\pm ABC}W^{\pm}_{[ABC]}\right\} \\
&= Ee^{-2\Phi}\left\{H_{ABC}H^{ABC} \pm 6H^{ABC}W^{\pm}_{ABC} + 3W^{\pm ABC}W^{\pm}_{ABC} - 6W^{\pm ABC}W^{\pm}_{BAC}\right\}. \tag{44}
\end{aligned}$$

Next, we calculate $Ee^{-2\Phi}T^A T_A$.

$$\begin{aligned}
&4Ee^{-2\Phi}T^A T_A \\
&= Ee^{-2\Phi}G^{MN}\left(2\partial_M\Phi - E^{KA}W^{\pm}_{KAB}E^B{}_M\right)\left(2\partial_N\Phi - E^{LC}W^{\pm}_{LCD}E^D{}_N\right) \\
&= Ee^{-2\Phi}\left\{4\partial^A\Phi\partial_A\Phi + W^{\pm A}{}_{AC}W^{\pm B}{}_B{}^C\right\} - 2E\partial_M(e^{-2\Phi})E^{MA}E^{NB}W^{\pm}_{NAB} \\
&= Ee^{-2\Phi}\left\{4\partial^A\Phi\partial_A\Phi + W^{\pm A}{}_{AC}W^{\pm B}{}_B{}^C + 2E^{MA}E^{NB}\partial_M W^{\pm}_{NAB}\right. \\
&\qquad\left. + 2E^K{}_C(\partial_M E^C{}_K)E^{MA}E^{NB}W^{\pm}_{NAB} + 2(\partial_M E^{MA})E^{NB}W^{\pm}_{NAB}\right. \\
&\qquad\left. + 2E^{MA}(\partial_M E^{NB})W^{\pm}_{NAB}\right\} - 2\partial_M\left(Ee^{-2\Phi}E^{MA}E^{NB}W^{\pm}_{NAB}\right) \\
&= Ee^{-2\Phi}\left\{4\partial^A\Phi\partial_A\Phi + W^{\pm A}{}_{AC}W^{\pm B}{}_B{}^C\right. \\
&\qquad\left. + E^{MA}E^{NB}\left(R^{\pm}_{MNAB} - W^{\pm}_{MA}{}^C W^{\pm}_{NCB} + W^{\pm}_{NA}{}^C W^{\pm}_{MCB}\right)\right. \\
&\qquad\left. - 2\Omega^A{}_A{}^C W^{\pm B}{}_{BC} + 2\Omega^{ACB}W^{\pm}_{CAB}\right\} + 2\partial_M\left(Ee^{-2\Phi}W^{\pm A}{}_A{}^M\right) \\
&= Ee^{-2\Phi}\left\{4\partial^A\Phi\partial_A\Phi + R^{\pm} \mp H^{ABC}W^{\pm}_{ABC} + W^{\pm ABC}W^{\pm}_{BAC}\right\} \tag{45} \\
&\qquad + 2\partial_M\left(Ee^{-2\Phi}W^{\pm A}{}_A{}^M\right),
\end{aligned}$$

where

$$R^{\pm}_{ABMN} \equiv \partial_M W^{\pm}_{NAB} - \partial_N W^{\pm}_{MAB} + W^{\pm}_{MA}{}^C W^{\pm}_{NCB} - W^{\pm}_{NA}{}^C W^{\pm}_{MCB}. \tag{46}$$

In the fourth equality, we used

$$\begin{aligned}
\partial_M E^M{}_A + E^K{}_C(\partial_M E^C{}_K)E^M{}_A &= E^{NB}\Omega_{NBA} = \Omega^B{}_{BA}, \\
E^{MA}(\partial_M E^{NB})W^{\pm}_{NAB} &= -E^{MA}E^{NB}(\partial_M E^C{}_N)W^{\pm}_{CAB} = \Omega^{ACB}W^{\pm}_{CAB}. \tag{47}
\end{aligned}$$

Now, we require invariance under general coordinate transformation. This means that $HW^{\pm}$ and $W^{\pm 2}$ terms should be removed by combining $Ee^{-2\Phi}S^{\pm}_{ABC}S^{\pm ABC}$, $Ee^{-2\Phi}T_A T^A$ and $Ee^{-2\Phi}W^{\pm ABC}W^{\pm}_{ABC}$. This uniquely constrains the form of the combination up to the overall factor, and the result becomes



$$Ee^{-2\Phi}\left(\frac{1}{6}S^{ABC}S_{ABC} + 4T^A T_A - \frac{1}{2}W^{\pm ABC}W^{\pm}_{ABC}\right)$$

$$= Ee^{-2\Phi}\left(4\partial^A\Phi\partial_A\Phi + R^{\pm} + \frac{1}{6}H^{ABC}H_{ABC}\right) + 2\partial_M\left(Ee^{-2\Phi}W^{\pm A}{}_A{}^M\right)$$

$$= Ee^{-2\Phi}\left(4\partial^A\Phi\partial_A\Phi + R - \frac{1}{2\cdot 3!}H^{ABC}H_{ABC}\right) + 2\partial_M\left(Ee^{-2\Phi}W^{\pm A}{}_A{}^M\right). \tag{48}$$

Thus, we construct NS-NS bosonic terms of the type II supergravities via $O(d) \times O(d)$ duality invariants. The Lagrangian is $O(d,d)$ invariant since it behaves as a scalar under general coordinate transformation and invariant under a constant shift of B field. Notice that the dual theory for + mode is written in terms of $E'^M_{(+)A}$, but it is possible to use local Lorentz transformation $E'^M_{(+)A} = E'^M_{(-)B}\Lambda^B{}_A$ to write the + mode of the dual theory in terms of $E'^M_{(-)A}$.

## 5. Construction of Fermionic Bilinear Terms in Type II Supergravities via Duality Invariants

Let us construct fermionic bilinear terms in the type II supergravities by using duality invariants. First, we consider bilinear terms of the dilatinos. Since the duality invariant forms of the dilatinos are given by $\Theta_{\pm}$, we would like to construct duality invariants that partially contain

$$\overline{\Theta}_{\pm}\Gamma^M\partial_M\Theta_{\pm}. \tag{49}$$

These are not duality invariants nor scalars under local Lorentz transformation. In order to recover the latter covariance, we add the connection $S^{\pm}_{ABC}$ as follows.

$$\overline{\Theta}_{\pm}\Gamma^M\partial_M\Theta_{\pm} \pm \frac{1}{12}\overline{\Theta}_{\pm}\Gamma^{ABC}S^{\pm}_{ABC}\Theta_{\pm}$$

$$= \overline{\Theta}_{\pm}\Gamma^M\left(\partial_M + \frac{1}{4}W^{\pm}_{MAB}\Gamma^{AB}\right)\Theta_{\pm} \pm \frac{1}{12}\overline{\Theta}_{\pm}\Gamma^{ABC}H_{ABC}\Theta_{\pm}$$

$$= \overline{\Theta}_{\pm}\Gamma^M D_M\Theta_{\pm} \mp \frac{1}{24}\overline{\Theta}_{\pm}\Gamma^{ABC}H_{ABC}\Theta_{\pm}. \tag{50}$$

Here, we used $\overline{\Theta}_{\pm}\Gamma^A\Theta_{\pm} = 0$ for Majorana fermions, and $D_M$ is a covariant derivative with respect to the connection of $\Omega_{MAB}$. In this case, $D_M = \partial_M + \frac{1}{4}\Omega_{MAB}\Gamma^{AB}$. Thus, the terms in Equation (50) are scalars under local Lorentz transformation. Furthermore, these are $O(d) \times O(d)$ duality invariant, as we show below. The dual theory is written by $E'^M_{(-)A}$ for the vielbein, and the dual of the above is written as

$$\overline{\Theta'}_{(-)\pm}\Gamma'^M_{(-)}\partial_M\Theta'_{(-)\pm} \pm \frac{1}{12}\overline{\Theta'}_{(-)\pm}\Gamma^{ABC}S'^{\pm}_{(-)ABC}\Theta'_{(-)\pm}$$

$$= \overline{\Theta'}_{(-)\pm}\Gamma'^M_{(-)}\left(\partial_M + \frac{1}{4}W'^{\pm}_{(-)MAB}\Gamma^{AB}\right)\Theta'_{(-)\pm} \pm \frac{1}{12}\overline{\Theta'}_{(-)\pm}\Gamma^{ABC}H'_{(-)ABC}\Theta'_{(-)\pm}$$

$$= \overline{\Theta}_{\pm}U_{\pm}^{-1}\Gamma'^M_{(-)}\left(\partial_M + \frac{1}{4}W'^{\pm}_{(-)MAB}\Gamma^{AB}\right)U_{\pm}\Theta_{\pm} \pm \frac{1}{12}\overline{\Theta}_{\pm}U_{\pm}^{-1}\Gamma^{ABC}H'_{(-)ABC}U_{\pm}\Theta_{\pm}$$

$$= \overline{\Theta}_{\pm}\Gamma'^M_{(\pm)}\left(\partial_M + \frac{1}{4}W'^{\pm}_{(\pm)MAB}\Gamma^{AB}\right)\Theta_{\pm} \pm \frac{1}{12}\overline{\Theta}_{\pm}\Gamma^{ABC}H'_{(\pm)ABC}\Theta_{\pm}$$

$$= \overline{\Theta}_{\pm}\Gamma'^M_{(\pm)}\partial_M\Theta_{\pm} \pm \frac{1}{12}\overline{\Theta}_{\pm}\Gamma^{ABC}S'^{\pm}_{(\pm)ABC}\Theta_{\pm}$$

$$= \overline{\Theta}_{\pm}\Gamma^M\partial_M\Theta_{\pm} \pm \frac{1}{12}\overline{\Theta}_{\pm}\Gamma^{ABC}S^{\pm}_{ABC}\Theta_{\pm}. \tag{51}$$

In the third equality, we used local Lorentz covariance for the + mode, such as $U_+^{-1}\Gamma'^M_{(-)}U_+ = \Gamma'^M_{(+)}$. Thus, the terms of Equation (50) are $O(d,d)$ invariant.

Next, we consider two derivative terms, which consist of $\Psi_{\pm M}$ and $\Theta_\pm$. The duality invariants should partially contain

$$\overline{\Psi}_{\pm M} G^{MN} \partial_N \Theta_\pm. \tag{52}$$

These are not duality invariants nor scalars under local Lorentz transformation. In order to make scalars under local Lorentz transformation, we need to add the connection term to the above.

$$\begin{aligned}
&\overline{\Psi}_{\pm M} G^{MN} \Big( \partial_N + \frac{1}{4} W^\pm_{NAB} \Gamma^{AB} \Big) \Theta_\pm \\
&= \overline{\Psi}^M_\pm D_M \Theta_\pm \mp \frac{1}{8} \overline{\Psi}^M_\pm H_{MAB} \Gamma^{AB} \Theta_\pm.
\end{aligned} \tag{53}$$

Furthermore, these are duality invariants, as we show below.

$$\begin{aligned}
&\overline{\Psi}'_{\pm M} G'^{MN} \Big( \partial_N + \frac{1}{4} W'^\pm_{(-)NAB} \Gamma^{AB} \Big) \Theta'_{(-)\pm} \\
&= \overline{\Psi}_{\pm M} G^{MK} Q^N_{\mp K} U^{-1}_\pm \Big( \partial_N + \frac{1}{4} W'^\pm_{(-)NAB} \Gamma^{AB} \Big) U_\pm \Theta_\pm \\
&= \overline{\Psi}_{\pm M} G^{MK} Q^N_{\mp K} \Big( \partial_N + \frac{1}{4} W'^\pm_{(\pm)NAB} \Gamma^{AB} \Big) \Theta_\pm \\
&= \overline{\Psi}_{\pm M} G^{MN} \Big( \partial_N + \frac{1}{4} W^\pm_{NAB} \Gamma^{AB} \Big) \Theta_\pm.
\end{aligned} \tag{54}$$

Thus, the terms of Equation (53) are $O(d,d)$ invariant.

Finally, let us investigate two derivative terms that are bilinear of Majorana gravitinos. These should partially contain the following terms.

$$\overline{\Psi}_{\pm L} G^{LN} \Gamma^M \partial_M \Psi_{\pm N}. \tag{55}$$

These are not duality invariants nor scalars under local Lorentz transformation. In order to recover the latter covariance, we add connection terms $S^\pm_{ABC}$ and $\Gamma^{\mp K}{}_{MN}$ as follows.

$$\begin{aligned}
&\overline{\Psi}_{\pm L} G^{LN} \Gamma^M \partial_M \Psi_{\pm N} \pm \frac{1}{12} \overline{\Psi}_{\pm L} G^{LN} \Gamma^{ABC} S^\pm_{ABC} \Psi_{\pm N} - \overline{\Psi}_{\pm L} G^{LN} \Gamma^M \Gamma^{\mp K}{}_{MN} \Psi_{\pm K} \\
&= \overline{\Psi}_{\pm L} G^{LN} \Gamma^M \Big( \partial_M + \frac{1}{4} W^\pm_{MAB} \Gamma^{AB} \Big) \Psi_{\pm N} \pm \frac{1}{12} \overline{\Psi}_{\pm L} G^{LN} \Gamma^{ABC} H_{ABC} \Psi_{\pm N} \\
&\quad - \overline{\Psi}_{\pm L} G^{LN} \Gamma^M \Gamma^{\mp K}{}_{MN} \Psi_{\pm K} \\
&= \overline{\Psi}^N_\pm \Gamma^M D_M \Psi_{\pm N} \mp \frac{1}{24} \overline{\Psi}^N_\pm \Gamma^{ABC} H_{ABC} \Psi_{\pm N} \mp \frac{1}{2} \overline{\Psi}_{\pm N} \Gamma_M H^{NMK} \Psi_{\pm K}.
\end{aligned} \tag{56}$$

Note that $D_M \Psi_{\pm N} = (\partial_M + \frac{1}{4} \Omega_{MAB} \Gamma^{AB}) \Psi_{\pm N} - \Gamma^K{}_{MN} \Psi_{\pm K}$. These are scalars under local Lorentz transformation. The first two terms are similar to Equation (51), so the transformations under $O(d) \times O(d)$ are also similar. One difference is in the derivative of $Q^{-1}_\mp$, which is written as

$$\begin{aligned}
&\overline{\Psi}'_{\pm L} G'^{LN} \Gamma'^M_{(-)} \partial_M \Psi'_{\pm N} \pm \frac{1}{12} \overline{\Psi}'_{\pm L} G'^{LN} \Gamma^{ABC} S'^\pm_{(-)ABC} \Psi'_{\pm N} \\
&= \overline{\Psi}_{\pm L} G^{LN} \Gamma^M \partial_M \Psi_{\pm N} \pm \frac{1}{12} \overline{\Psi}_{\pm L} G^{LN} \Gamma^{ABC} S^\pm_{ABC} \Psi_{\pm N} \\
&\quad + (\partial_M Q^{-1N'}_\mp{}_N) Q^N_{\mp K} \overline{\Psi}_{\pm L} G^{LK} \Gamma^M \Psi_{\pm N'}.
\end{aligned} \tag{57}$$

On the other hand, the $O(d) \times O(d)$ transformations of the connections $\Gamma^{\mp K}{}_{MN}$ are given by Equation (18), and the third term in Equation (56) transforms as

$$
\begin{aligned}
&- \overline{\Psi}'_{\pm L} G'^{LN} \Gamma'^{M}_{(-)} \Gamma'^{\mp K}{}_{MN} \Psi'_{\pm K} \\
&= - \overline{\Psi}_{\pm L} G^{LN'} Q^{N}_{\mp N'} U^{-1}_{\pm} \Gamma'^{M}_{(-)} U_{\pm} \Gamma'^{\mp K}{}_{MN} \Psi_{\pm K'} Q^{-1 K'}_{\mp}{}_{K} \\
&= - \overline{\Psi}_{\pm L} G^{LN'} \Gamma^{M'} Q^{-1 K'}_{\mp}{}_{K} \Gamma'^{\mp K}{}_{MN} Q^{M}_{\pm M'} Q^{N}_{\mp N'} \Psi_{\pm K'} \\
&= - \overline{\Psi}_{\pm L} G^{LN} \Gamma^{M} \Gamma^{\mp K}{}_{MN} \Psi_{\pm K} + Q^{-1 K'}_{\mp}{}_{K} (\partial_{M} Q^{K}_{\mp N}) \overline{\Psi}_{\pm L} G^{LN} \Gamma^{M} \Psi_{\pm K'}.
\end{aligned}
\tag{58}
$$

In the second equality, we used $U^{-1}_{\pm} \Gamma'^{M}_{(-)} U_{\pm} = \Gamma'^{M}_{(\pm)} = Q^{M}_{\pm M'} \Gamma^{M'}$. Thus, we see that the last term in Equation (57) is cancelled by the last term in Equation (58). The combinations of Equation (56) are $O(d,d)$ invariant.

So far, we constructed $O(d,d)$ invariants of (50), (53) and (56). Then, up to overall factor, the Lagrangian is expressed as

$$
\begin{aligned}
E e^{-2\Phi} &\Big[ \overline{\Theta}_{\pm} \Gamma^{M} D_{M} \Theta_{\pm} \mp \frac{1}{24} \overline{\Theta}_{\pm} \Gamma^{ABC} H_{ABC} \Theta_{\pm} + c_1 \Big\{ \overline{\Psi}^{M}_{\pm} D_{M} \Theta_{\pm} \mp \frac{1}{8} \overline{\Psi}^{M}_{\pm} H_{MAB} \Gamma^{AB} \Theta_{\pm} \Big\} \\
&+ c_2 \Big\{ \overline{\Psi}^{N}_{\pm} \Gamma^{M} D_{M} \Psi_{\pm N} \mp \frac{1}{24} \overline{\Psi}^{N}_{\pm} \Gamma^{ABC} H_{ABC} \Psi_{\pm N} \mp \frac{1}{2} \overline{\Psi}_{\pm N} \Gamma_{M} H^{NMK} \Psi_{\pm K} \Big\} \Big]
\end{aligned}
$$

$$
\begin{aligned}
= E e^{-2\Phi} &\Big[ \overline{\lambda}_{\pm} \Gamma^{M} D_{M} \lambda_{\pm} - \overline{\lambda}_{\pm} \Gamma^{M} D_{M} (\Gamma^{A} \Psi_{\pm A}) + \overline{\Psi}_{\pm A} \Gamma^{A} \Gamma^{M} D_{M} \lambda_{\pm} - \overline{\Psi}_{\pm A} \Gamma^{A} \Gamma^{M} D_{M} (\Gamma^{B} \Psi_{\pm B}) \\
&\mp \frac{1}{24} \overline{\lambda}_{\pm} \Gamma^{ABC} H_{ABC} \lambda_{\pm} \mp \frac{1}{12} \overline{\Psi}_{\pm D} \Gamma^{D} \Gamma^{ABC} H_{ABC} \lambda_{\pm} \pm \frac{1}{24} \overline{\Psi}_{\pm D} \Gamma^{D} \Gamma^{ABC} \Gamma^{E} H_{ABC} \Psi_{\pm E} \\
&+ c_1 \Big\{ \overline{\Psi}^{M}_{\pm} D_{M} \lambda_{\pm} - \overline{\Psi}^{M}_{\pm} D_{M} (\Gamma^{A} \Psi_{\pm A}) \mp \frac{1}{8} \overline{\Psi}^{M}_{\pm} H_{MAB} \Gamma^{AB} \lambda_{\pm} \pm \frac{1}{8} \overline{\Psi}^{M}_{\pm} H_{MAB} \Gamma^{AB} \Gamma^{C} \Psi_{\pm C} \Big\} \\
&+ c_2 \Big\{ \overline{\Psi}^{N}_{\pm} \Gamma^{M} D_{M} \Psi_{\pm N} \mp \frac{1}{24} \overline{\Psi}^{N}_{\pm} \Gamma^{ABC} H_{ABC} \Psi_{\pm N} \mp \frac{1}{2} \overline{\Psi}_{\pm N} \Gamma_{M} H^{NMK} \Psi_{\pm K} \Big\} \Big]
\end{aligned}
$$

$$
\begin{aligned}
= E e^{-2\Phi} &\Big[ \overline{\lambda}_{\pm} \Gamma^{M} D_{M} \lambda_{\pm} - \overline{\lambda}_{\pm} \Gamma^{M} \Gamma^{A} D_{M} \Psi_{\pm A} - \overline{\Psi}_{\pm A} \Gamma^{M} \Gamma^{A} D_{M} \lambda_{\pm} + \overline{\Psi}_{\pm A} \Gamma^{M} \Gamma^{AB} D_{M} \Psi_{\pm B} \\
&\mp \frac{1}{24} \overline{\lambda}_{\pm} \Gamma^{ABC} H_{ABC} \lambda_{\pm} \mp \frac{1}{12} \overline{\Psi}_{\pm D} \Gamma^{DABC} H_{ABC} \lambda_{\pm} \pm \frac{1}{24} \overline{\Psi}_{\pm D} \Gamma^{DABCE} H_{ABC} \Psi_{\pm E} \\
&+ (2 + c_1) \overline{\Psi}^{M}_{\pm} D_{M} \lambda_{\pm} - (2 + c_1) \overline{\Psi}^{M}_{\pm} \Gamma^{A} D_{M} \Psi_{\pm A} + (1 + c_2) \overline{\Psi}^{A}_{\pm} \Gamma^{M} D_{M} \Psi_{\pm A} \\
&\mp \frac{2 + c_1}{8} \overline{\Psi}^{M}_{\pm} H_{MAB} \Gamma^{AB} \lambda_{\pm} \pm \frac{2 + c_1}{8} \overline{\Psi}^{M}_{\pm} H_{MAB} \Gamma^{AB} \Gamma^{C} \Psi_{\pm C} \\
&\mp \frac{1 + c_2}{24} \overline{\Psi}^{N}_{\pm} \Gamma^{ABC} H_{ABC} \Psi_{\pm N} \mp \frac{1 + 2c_2}{4} \overline{\Psi}_{\pm N} \Gamma_{M} H^{NMK} \Psi_{\pm K} \Big].
\end{aligned}
\tag{59}
$$

In the last equality, if we choose $c_1 = -2$ and $c_2 = -1$, it is possible to express the derivative of the Majorana gravitinos as field strengths of $D_{[M} \Psi_{\pm N]}$ up to partial integral. Since this prescription is important to realize local supersymmetry, we employ these values. Then, the $O(d,d)$ invariant action of the fermionic bilinear is uniquely determined as

$$
\begin{aligned}
E e^{-2\Phi} &\Big[ \overline{\Theta}_{\pm} \Gamma^{M} D_{M} \Theta_{\pm} \mp \frac{1}{24} \overline{\Theta}_{\pm} \Gamma^{ABC} H_{ABC} \Theta_{\pm} - 2 \Big\{ \overline{\Psi}^{M}_{\pm} D_{M} \Theta_{\pm} \mp \frac{1}{8} \overline{\Psi}^{M}_{\pm} H_{MAB} \Gamma^{AB} \Theta_{\pm} \Big\} \\
&- \Big\{ \overline{\Psi}^{N}_{\pm} \Gamma^{M} D_{M} \Psi_{\pm N} \mp \frac{1}{24} \overline{\Psi}^{N}_{\pm} \Gamma^{ABC} H_{ABC} \Psi_{\pm N} \mp \frac{1}{2} \overline{\Psi}_{\pm N} \Gamma_{M} H^{NMK} \Psi_{\pm K} \Big\} \Big]
\end{aligned}
$$

$$
\begin{aligned}
= E e^{-2\Phi} &\Big[ \overline{\lambda}_{\pm} \Gamma^{M} D_{M} \lambda_{\pm} - \overline{\lambda}_{\pm} \Gamma^{M} \Gamma^{A} D_{M} \Psi_{\pm A} - \overline{\Psi}_{\pm A} \Gamma^{M} \Gamma^{A} D_{M} \lambda_{\pm} + \overline{\Psi}_{\pm A} \Gamma^{M} \Gamma^{AB} D_{M} \Psi_{\pm B} \\
&\mp \frac{1}{24} \overline{\lambda}_{\pm} \Gamma^{ABC} H_{ABC} \lambda_{\pm} \mp \frac{1}{12} \overline{\Psi}_{\pm D} \Gamma^{DABC} H_{ABC} \lambda_{\pm} \pm \frac{1}{24} \overline{\Psi}_{\pm D} \Gamma^{DABCE} H_{ABC} \Psi_{\pm E} \\
&\pm \frac{1}{4} \overline{\Psi}_{\pm N} \Gamma_{M} H^{NMK} \Psi_{\pm K} \Big].
\end{aligned}
\tag{60}
$$

Of course, a linear combination of these terms is consistent with the type II supergravities. Thus, we showed that fermionic bilinears without R-R fields can be written in terms of the duality invariants within the framework of the type II supergravities. Invariant forms of fermionic bilinears with R-R fluxes are obtained in the framework of the double field theory [31] or generalized geometry [35].

## 6. Conclusions and Discussion

In this paper, within the framework of the type II supergravities, we have constructed $O(d) \times O(d)$ duality invariants of Equations (31), (33) and (35) by examining $O(d) \times O(d)$ transformations of three-form H field, dilaton and dilatino. These invariants are checked in the background of fundamental strings and wave solutions, or NS5-branes and KK monopoles. By using these duality invariants, we reconstructed the actions of type II supergravities in a manifestly $O(d) \times O(d)$ invariant form in Sections 4 and 5. Since these actions are also invariant under linear $GL(d)$ transformation and shift of the B field, these are exactly $O(d, d)$ invariant. As for the kinetic terms on R-R fields, $SO(d, d)$ invariant construction was already discussed within the framework of the type II supergravities in ref. [20].

As we have checked the duality invariants in the background of strings and wave solutions, or NS5-branes and KK monopoles, it is easy to apply to other non-geometric backgrounds [37–40]. It is interesting to see corrections to the non-geometric background, which was studied from the viewpoint of world-sheet instantons [41]. It is also interesting to investigate $\beta$-twisted solutions of the double field theory [42] by evaluating $O(d) \times O(d)$ invariants in this paper.

Since we have constructed $O(d) \times O(d)$ duality invariants within the framework of the type II supergravities, it is natural to generalize these formulations to higher derivative corrections in the type II superstring theories. However, this is not a simple task, and it is shown that higher derivative corrections in bosonic or heterotic string theory cannot be written in terms of generalized metric [43,44]. We should take into account total derivative terms and field redefinitions, which consist of dimensionally reduced fields. Constraint on $R^2$ terms via cosmological ansatz was investigated in ref. [45], was executed via T-duality in refs. [46] and was performed via $O(d, d)$ duality in ref. [47]. In our formalism, the difficulty can be seen by duality transformation of the Riemann tensor (46), which is calculated as

$$
\begin{aligned}
R'^{\pm}_{(\pm)ABCD} &= E'^{M}_{(\pm)C} E'^{N}_{(\pm)D} R'^{\pm}_{(\pm)ABMN} \\
&= R^{\pm}_{ABCD} \pm 2 R^{\pm}_{ABN[C} X^{N}_{\mp D]} + 2 X^{M}_{\mp[C} X^{N}_{\mp D]} W^{\pm}_{MAE} W^{\pm E}_{N\ B} \\
&\quad \mp 2 W^{\pm}_{KAB} W^{\pm}_{[CD]E} X^{KE}_{\mp} + 2 W^{\pm}_{KAB} W^{\pm}_{LE[C} X^{L}_{\mp D]} X^{KE}_{\mp} \\
&\quad + X^{KE}_{\mp} S_{ECD} W^{\pm}_{KAB},
\end{aligned}
\tag{61}
$$

where $X^{KE}_{\mp} = Q^{-1K}_{\mp}{}_{L}(S - R)^{LE} = -X^{EK}_{\pm}$. If we consider $R^{\pm}_{ABCD} S^{\pm ABE} S^{\pm CD}{}_{E}$, which exists as a part of higher derivative terms in bosonic string theory, the duality transformation of this term contains $\pm 2 R^{\pm}_{ABN[C} X^{N}_{\mp D]} S^{\pm ABE} S^{\pm CD}{}_{E}$. However, this cannot be cancelled by other terms even if we consider total derivatives and field redefinitions of 10 dimensional fields.

Although we should decompose Equation (61) in terms of dimensionally reduced fields, $S_{ABC}$, $T_A$ and $W^{\pm}_{\mu AB}$ are invariant under $O(d, d)$ transformations. Thus, we should only take care of $W^{\pm}_{\alpha AB}$. It is also useful to consult a frame formalism of the double field theory [48]. If we find nice structure on total derivatives and field redefinitions in terms of these fields, it will be possible to apply our $O(d) \times O(d)$ construction to higher derivative terms such as $R^4$ terms [49–55].

**Author Contributions:** Conceptualization, Y.H. and K.M.; methodology, Y.H. and K.M.; formal analysis, K.M.; investigation, Y.H.; resources, Y.H.; data curation, K.M.; writing—original draft preparation, Y.H. and K.M.; writing—review and editing, Y.H. and K.M.; supervision, Y.H.; funding acquisition, Y.H. All authors have read and agreed to the published version of the manuscript.

**Funding:** This work was partially supported by the Japan Society for the Promotion of Science, Grant-in-Aid for Scientific Research (C) Grant Number 22K03613.

**Data Availability Statement:** Data are contained within the article.

**Acknowledgments:** The authors would like to thank Takanori Fujiwara and Makoto Sakaguchi. Y.H. would like to thank Tetsuji Kimura and Kentaroh Yoshida for useful conversations. We would also like to thank Yuho Sakatani for valuable comments.

**Conflicts of Interest:** The authors declare no conflicts of interest.

## Appendix A. Review of Type II Supergravities

The type II supergravities consist of massless fields of corresponding type II superstring theories. Bosonic massless fields are the graviton $G_{MN}$, the dilaton $\Phi$, NS-NS B field $B_{MN}$ and R-R fields, but in this appendix we ignore R-R fields of the type II supergravities. Fermionic massless fields are two gravitinos $\Psi_{\pm M}$ and two dilatinos $\lambda_\pm$. Physical degrees of freedom for bosonic fields and fermionic ones are 128, so it is possible to relate bosonic and fermionic fields via local supersymmetry.

Historically, the Lagrangian of the type IIA supergravity is derived by the dimensional reduction of 11 dimensional supergravity into 10 dimensions, and its explicit form is given by [4]

$$
\begin{aligned}
\mathcal{L} = Ee^{-2\Phi}\Big[ & 4\partial^A\Phi\partial_A\Phi + R - \frac{1}{2\cdot 3!}H^{ABC}H_{ABC} \\
& + \bar{\lambda}\Gamma^M D_M\lambda - \bar{\lambda}\Gamma^M\Gamma^A D_M\Psi_A - \overline{\Psi}_A\Gamma^M\Gamma^A D_M\lambda + \overline{\Psi}_A\Gamma^M\Gamma^{AB}D_M\Psi_B \\
& - \frac{1}{24}\bar{\lambda}\Gamma^{ABC}H_{ABC}\Gamma_{11}\lambda - \frac{1}{12}\overline{\Psi}_D\Gamma^{DABC}H_{ABC}\Gamma_{11}\lambda - \frac{1}{24}\overline{\Psi}_D\Gamma^{DABCE}H_{ABC}\Gamma_{11}\Psi_E \\
& - \frac{1}{4}\overline{\Psi}_N\Gamma_M H^{NMK}\Gamma_{11}\Psi_K \Big] \\
= Ee^{-2\Phi}\Big[ & 4\partial^A\Phi\partial_A\Phi + R - \frac{1}{2\cdot 3!}H^{ABC}H_{ABC} \\
& + \bar{\lambda}_+\Gamma^M D_M\lambda_+ - \bar{\lambda}_+\Gamma^M\Gamma^A D_M\Psi_{+A} - \overline{\Psi}_{+A}\Gamma^M\Gamma^A D_M\lambda_+ + \overline{\Psi}_{+A}\Gamma^M\Gamma^{AB}D_M\Psi_{+B} \\
& - \frac{1}{24}\bar{\lambda}_+\Gamma^{ABC}H_{ABC}\lambda_+ - \frac{1}{12}\overline{\Psi}_{+D}\Gamma^{DABC}H_{ABC}\lambda_+ + \frac{1}{24}\overline{\Psi}_{+D}\Gamma^{DABCE}H_{ABC}\Psi_{+E} \\
& + \frac{1}{4}\overline{\Psi}_{+N}\Gamma_M H^{NMK}\Psi_{+K} \\
& + \bar{\lambda}_-\Gamma^M D_M\lambda_- - \bar{\lambda}_-\Gamma^M\Gamma^A D_M\Psi_{-A} - \overline{\Psi}_{-A}\Gamma^M\Gamma^A D_M\lambda_- + \overline{\Psi}_{-A}\Gamma^M\Gamma^{AB}D_M\Psi_{-B} \\
& + \frac{1}{24}\bar{\lambda}_-\Gamma^{ABC}H_{ABC}\lambda_- + \frac{1}{12}\overline{\Psi}_{-D}\Gamma^{DABC}H_{ABC}\lambda_- - \frac{1}{24}\overline{\Psi}_{-D}\Gamma^{DABCE}H_{ABC}\Psi_{-E} \\
& - \frac{1}{4}\overline{\Psi}_{-N}\Gamma_M H^{NMK}\Psi_{-K} \Big].
\end{aligned}
\tag{A1}
$$

In the above, we neglected quartic terms on fermionic fields. Here, $\lambda$ and $\Psi_M$ are Majorana fermions and satisfy

$$
\begin{aligned}
\lambda = \lambda_+ + \lambda_-, \quad & \Psi_M = \Psi_{+M} + \Psi_{-M}, \\
\Gamma_{11}\lambda_\pm = \pm\lambda_\pm, \quad & \Gamma_{11}\Psi_{\pm M} = \mp\Psi_{\pm M}.
\end{aligned}
\tag{A2}
$$

We chose similar notations as in ref. [56,57]. The Lagrangian of the type IIB supergravity takes a similar form as Equation (A1), but $\pm$ modes of the dilatinos or the gravitinos have the same chirality.

$$
\Gamma_{11}\lambda_\pm = -\lambda_\pm, \quad \Gamma_{11}\Psi_{\pm M} = \Psi_{\pm M}.
\tag{A3}
$$

In the case of the type IIA supergravity, transformations of massless fields under local supersymmetry are given by

$$\delta E^A{}_M = \bar{\epsilon}\Gamma^A\Psi_M = \bar{\epsilon}_+\Gamma^A\Psi_{+M} + \bar{\epsilon}_-\Gamma^A\Psi_{-M},$$
$$\delta\Phi = \frac{1}{2}\bar{\epsilon}\lambda = \frac{1}{2}\bar{\epsilon}_+\lambda_- + \frac{1}{2}\bar{\epsilon}_-\lambda_+,$$
$$\delta B_{MN} = 2\bar{\epsilon}\Gamma_{11}\Gamma_{[M}\Psi_{N]} = 2\bar{\epsilon}_+\Gamma_{[M}\Psi_{+N]} - 2\bar{\epsilon}_-\Gamma_{[M}\Psi_{-N]},$$
$$\delta\lambda = 2(\partial_M\Phi)\Gamma^M\epsilon + \frac{1}{6}H_{ABC}\Gamma^{ABC}\Gamma_{11}\epsilon$$
$$= 2\Big\{(\partial_M\Phi)\Gamma^M - \frac{1}{12}H_{ABC}\Gamma^{ABC}\Big\}\epsilon_+ + 2\Big\{(\partial_M\Phi)\Gamma^M + \frac{1}{12}H_{ABC}\Gamma^{ABC}\Big\}\epsilon_-,$$
$$\delta\Psi_M = 2D_M\epsilon + \frac{1}{4}H_{MAB}\Gamma^{AB}\Gamma_{11}\epsilon$$
$$= 2\Big(\partial_M + \frac{1}{4}W^+_{MAB}\Gamma^{AB}\Big)\epsilon_+ + 2\Big(\partial_M + \frac{1}{4}W^-_{MAB}\Gamma^{AB}\Big)\epsilon_-. \tag{A4}$$

Again, we ignored contributions of R-R fields and cubic terms on fermionic fields. $\epsilon$ is a Majorana fermion and satisfies

$$\epsilon = \epsilon_+ + \epsilon_-, \quad \Gamma_{11}\epsilon_\pm = \mp\epsilon_\pm. \tag{A5}$$

In the case of the type IIB supergravity, $\epsilon_\pm$ should satisfy

$$\Gamma_{11}\epsilon_\pm = \epsilon_\pm. \tag{A6}$$

## Note

[1]     We neglect R-R fields since these are already completed in ref. [20].

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
