# Peer review of "Reconstruction of Type II Supergravities via O(d) × O(d) Duality Invariants"

_universe, doi:10.3390/universe10010028_

Round 1

Reviewer 1 Report

Comments and Suggestions for Authors

The authors construct invariants of the subgroup O(d) x O(d) of the T-duality group O(d, d) of ten-dimensional type-II supergravity. They then show that starting from their invariants they can systematically reconstruct the bosonic NS-NS and bilinear fermionic terms of the supergravity action. These results build on previous work, and are consistent with results obtained by generalized geometry and double field theory.

Despite this partial overlap of results, I think the approach taken by the authors is interesting, because it directly works within the supergravity formalism without doubling indices (as in generalized geometry) or coordinates (as in double field theory). As the authors point out, there are important open questions regarding the action of T-duality on higher derivative terms. I think the quality and quantity of results clearly merit publication. 

The paper, while technical in nature, is written in a comprehensible way, which allows readers to understand, and, with some effort, check calculations. The references given are comprehensive. I do not have suggestions for substantial changes, but I have spotted two minor points which I would like to ask the authors to check. 

Formula (9), first line: is there an explicit expression for H’_{MNK}? In Ref. [20] I could only find the second line. 

Formula (13): Should Gamma on the left hand side have a `prime’?

Author Response

Dear referee,

Thank you for reviewing our paper.

> Formula (9), first line: is there an explicit expression for H’_{MNK}? 
> In Ref. [20] I could only find the second line.

We constructed duality transformation of H-field in local Lorentz frame, and did not use H’_{MNK}. It may be possible to omit the first line, but we respect the definition of H_{ABC}. If we want to obtain H'_{MNK}, we should combine the second line and transformations of the vielbein.

> Formula (13): Should Gamma on the left hand side have a `prime’?

Yes, we have added ‘prime’ to Gamma on the left hand side.

We also added some explanations as follows. (The equation number is changed.)
* In section 2, we added explanations on the dimensional reduction in l.74 - l.99.
  5 equations are added, and the eq.(7) is modified.
* In the 4th line of the eq.(22), we added \epsilon_-.
* In section 3.3, we added detailed explanations on classical solutions in l.119 - l.130.
* In appendix A, the basics of the type IIA supergravity is added in l.193 - l.198.

with best,
Yoshifumi Hyakutake

Reviewer 2 Report

Comments and Suggestions for Authors

Report of the Referee

Manuscript Ref.:  Universe-2781536

Title: "Reconstruction of Type II Supergravities via O(d)×O(d) Duality Invariants"

==========================================

The authors presented an interesting work on constructing  O(d) x O(d) duality invariants by examining O(d) x O(d) transformations of 3-form H field, dilaton and dilatino. This is done in the context of type II supergravity theories (type II STs). Using these duality they reconstructed the actions of type II STs in a manifestly O(d) x O(d)  invariant form.  In particular, fermionic bilinear terms are constructed in the type II supergravities by duality invariants. The work revisits some results on O(d) x O(d) subgroup of the  duality published in Ref. [20] in order  to construct the invariants within  the framework of the type II STs. Probably, the formalism can be extended to include other nongeometric backgrounds or higher derivative corrections in the type II STs.

The work is relatively well presented and understandable. The references are sufficiently updated. The topic is relevant for practical use in the area of phenomenology of quantum gravity and superstring theories. My naive question is about the possibility to put forward cosmological solutions based on the present formalism and in case of positive answer discuss their physical interpretations for simplified scenarios.

For these reasons, I recommend the manuscript for publication in its present form.

Author Response

Dear referee,

Thank you for reviewing our paper.

>My naive question is about the possibility to put forward cosmological  >solutions based on the present formalism and in case of positive answer discuss >their physical interpretations for simplified scenarios.

It is an interesting point and that is our initial motivation to this work. 
We hope to show some results in future. 

We also added some explanations as follows. (The equation number is changed.)
* In section 2, we added explanations on the dimensional reduction in l.74 - l.99.
  5 equations are added, and the eq.(7) is modified.
* In the eq.(18), we have added the prime to the Gamma on the left hand side.
* In the 4th line of the eq.(22), we added \epsilon_-.
* In section 3.3, we added detailed explanations on classical solutions in l.119 - l.130.
* In appendix A, the basics of the type IIA supergravity is added in l.193 - l.198.

with best,
Yoshifumi Hyakutake